# ICER is requisite for Th17 differentiation

Nobuya Yoshida[1], Denis Comte[1,2], Masayuki Mizui[1], Kotaro Otomo[1], Florencia Rosetti[3], Tanya N. Mayadas[3], José C. Crispín[1], Sean J. Bradley[1], Tomohiro Koga[1], Michihito Kono[1], Maria P. Karampetsou[1], Vasileios C. Kyttaris[1], Klaus Tenbrock[4] & George C. Tsokos[1]

Inducible cAMP early repressor (ICER) has been described as a transcriptional repressor isoform of the cAMP response element modulator (CREM). Here we report that ICER is predominantly expressed in Th17 cells through the IL-6–STAT3 pathway and binds to the *Il17a* promoter, where it facilitates the accumulation of the canonical enhancer RORγt. *In vitro* differentiation from naive ICER/CREM-deficient CD4$^+$ T cells to Th17 cells is impaired but can be rescued by forced overexpression of ICER. Consistent with a role of Th17 cells in autoimmune and inflammatory diseases, ICER/CREM-deficient B6.*lpr* mice are protected from developing autoimmunity. Similarly, both anti-glomerular basement membrane-induced glomerulonephritis and experimental encephalomyelitis are attenuated in ICER/CREM-deficient mice compared with their ICER/CREM-sufficient littermates. Importantly, we find ICER overexpressed in CD4$^+$ T cells from patients with systemic lupus erythematosus. Collectively, our findings identify a unique role for ICER, which affects both organ-specific and systemic autoimmunity in a Th17-dependent manner.

[1] Division of Rheumatology, Beth Israel Deaconess Medical Center, Harvard Medical School, Boston, Massachusetts 02215, USA. [2] Division of Immunology and Allergy, Centre Hospitalier Universitaire Vaudois, 1011 Lausanne, Switzerland. [3] Center for Excellence in Vascular Biology, Department of Pathology, Brigham and Women's Hospital, Harvard Medical School, Boston, Massachusetts 02215, USA. [4] Department of Pediatrics, Division of Allergology and Immunology, RWTH University of Aachen, 52056 Aachen, Germany. Correspondence and requests for materials should be addressed to N.Y. (email: nyoshida@bidmc.harvard.edu) or to G.C.T. (email: gtsokos@bidmc.harvard.edu).

The role of cAMP-response element modulator (CREM) in T-cell differentiation is complex and not completely understood. CREM has many alternatively spliced transcript variants, and their relative expression affects T-cell differentiation. A relationship between CREM and Th17 cells has been proposed[1]. Genome-wide analyses of Th17 transcription regulatory network revealed the induction of CREM among other genes, and silencing of *CREM* was associated with reduced Th17 differentiation[2].

Numerous reports have claimed that interleukin (IL)-17 has an important role in the pathogenesis of autoimmune diseases, including systemic lupus erythematosus (SLE)[1,3–5]. Expression of CREMα, a repressor isoform of CREM, is increased in CD4[+] T cells from SLE patients, and forced expression of CREMα in human T cells enhances IL-17A expression[6]. Moreover, mice overexpressing CREMα in T cells display increased IL-17 production and severe skin inflammation, as well as mild lupus-like disease[7].

Inducible cAMP early repressor (ICER) is a splice variant of CREM[8]. In contrast to other isoforms of *CREM*, *ICER* has an alternative transcription initiation site and is induced by a unique alternative promoter (P2)[9]. Because ICER has no transcriptional activation domains, it functions as a powerful repressor of cAMP-induced CRE-mediated transcription. Previous papers have shown that ICER inhibits T-cell activation, Th1/Th2 cell differentiation and suppresses the production of proinflammatory cytokines[10,11]; however, whether ICER is involved in the generation of Th17 cells is not known.

Here we demonstrate that ICER is the predominant CREM isoform expressed in Th17 cells in both mice and humans. ICER is induced by IL-6 via STAT3 signalling and enhances RORγt accumulation on the *Il17a* promoter. Mice deficient in ICER/CREM develop less anti-glomerular basement membrane-induced glomerulonephritis (AIGN) and experimental encephalomyelitis (EAE), and B6.lpr ICER/CREM-deficient mice develop less autoimmunity and lupus nephritis. The relevance of these new findings in human disease is underscored by the increased expression of ICER in T cells from SLE patients. Overall, ICER controls organ-specific and systemic autoimmunity by controlling IL-17 production.

## Results

**ICER is induced in Th17-polarized murine CD4 T cells.** To further understand the role of CREM in Th17 differentiation, we asked which CREM isoforms are expressed during *in vitro* Th17-polarizing conditions. Using western blotting we investigated the expression of CREM isoform induction in ICER/CREM-sufficient and -deficient T cells cultured under Th17-polarizing conditions. We noted a < 20 kDa CREM band to be induced by day 3 (Fig. 1a, left) in ICER/CREM-sufficient but not in ICER/CREM-deficient T cells (Fig. 1a, right first and second lanes) pointing that this band was an isoform of CREM. Because the size of the detected CREM was < 20 kDa, we assumed that it represented ICER. Because CREM has various isoforms and it is impossible to identify specific isoform(s) by conventional mass spectrometry, we generated plasmids that overexpress each of the two typical ICER isoforms, that is, ICER and ICERγ. When we transfected these plasmids into HEK-293T cells and compared the size of those molecules with the < 20 kDa CREM, both overexpressed ICER bands (majority ICERγ) fit perfectly to the < 20 kDa CREM (Fig. 1a, right). We conclude that the < 20 kDa CREM induced in Th17 cells is an isoform of ICERγ or perhaps ICER.

Next we polarized T cells under Th17, Th1, Th2 and Treg conditions, and we noted that ICER was present in significant amounts only when cells were driven towards Th17 rather than any of the other three conditions (Fig. 1b). To confirm that ICER is induced under Th17 conditions, we cultured T cells from B6.IL-17[GFP] mice under the same Th17 culture conditions and sorted green fluorescent protein (GFP)-positive (IL-17A producing) and GFP-negative (IL-17A non-producing) cells. As seen in Fig. 1c, ICER is expressed primarily by IL-17A-producing cells and not by non-IL-17A-producing cells (Fig. 1c). We asked whether IL-17A-producing T cells from MRL/*lpr* mice before *in vitro* polarization also express ICER as a result of ongoing stimulation *in vivo*. Indeed, as shown in Fig. 1d, IL-17A-producing T cells from MRL/*lpr* mice expressed more ICER than naive CD4 MRL/Mpj mice.

**ICER/CREM deficiency reduces *in vitro* Th17 differentiation.** Although ICER is known to affect Th1 and Th2 *in vitro* polarization, it remains unclear how ICER/CREM deficiency contributes to Th17 and Treg differentiation. We assessed the impact of ICER expression in T cells cultured *in vitro* under Th1, Th2, Th17 and Treg polarized conditions. As shown in Fig. 2a, IL-17A-producing cells were reduced in Th17-polarized cells from ICER/CREM-deficient compared with those from ICER/CREM-sufficient cells, yet, we could not see any significant differences in T cells cultured under other polarizing conditions. We determined the expression of each master transcription factor, *Gata3*, *Tbet*, *Rorc* and *Foxp3* in T cells cultured under Th1, Th2, Th17 and Treg, respectively. We did not find any differences in the expression of *Tbet* and *Foxp3* but we found increased expression of *Gata3* and decreased *Rorc* gene expression in ICER/CREM-deficient mice (Fig. 2b). These data are in agreement with a previous report according to which ICER/CREM-deficient T cells tend to differentiate towards Th2 more than ICER/CREM-sufficient T cells[11,12]. Decreased numbers of IL-17A-producing cells among Th17-polarized ICER/CREM-deficient T cells were conserved when several IL-6 concentrations were used (Fig. 2c). When we measured the expression of Th17-related genes, ICER/CREM-deficient T cells displayed less *Il23r*, *Il17a* and *Il17f* expression (Fig. 2d).

To prove further the role of ICER in Th17 differentiation we transfected ICER/CREM-deficient T cells with ICER and ICERγ expression vectors and polarized them under Th17 conditions. As seen in Fig. 2g, transfection of either of the ICER isoforms resulted in increased IL-17 GFP[+] T-cell induction relative to those that received empty plasmid, providing further support that ICER promotes Th17 differentiation. Taken together, these data indicate that ICER/CREM promotes Th17 differentiation, whereas it inhibits Th2 differentiation and does not affect Th1 and Treg differentiation.

**ICER is induced by IL-6 through STAT3 signalling.** Combination of IL-6 and transforming growth factor beta (TGFβ) can differentiate T cells into Th17 cells, where transduction of IL-6 receptor activity occurs through STAT3 and that of TGFβ through SMAD3 (refs 4,13). To address the relative impact of these stimuli we added to anti-CD3 and anti-CD28 Ab-stimulated T cells IL-6, TGFβ, or IL-6 and TGFβ. ICER was induced only if IL-6 was present in the culture (Fig. 3a). This finding was confirmed by blocking IL-6 or TGFβ signal transduction by adding STAT3 or SMAD3 inhibitors in T cells cultured under polarizing conditions. The presence of STAT3 inhibitor inhibited the induction of ICER in a dose-dependent manner while the presence of a SMAD3 inhibitor had no effect (Fig. 3b). Furthermore, when we cultured naive CD4[+] T cells from CD4[cre+] STAT3[fl/fl] mice under Th17-polarizing conditions we could not

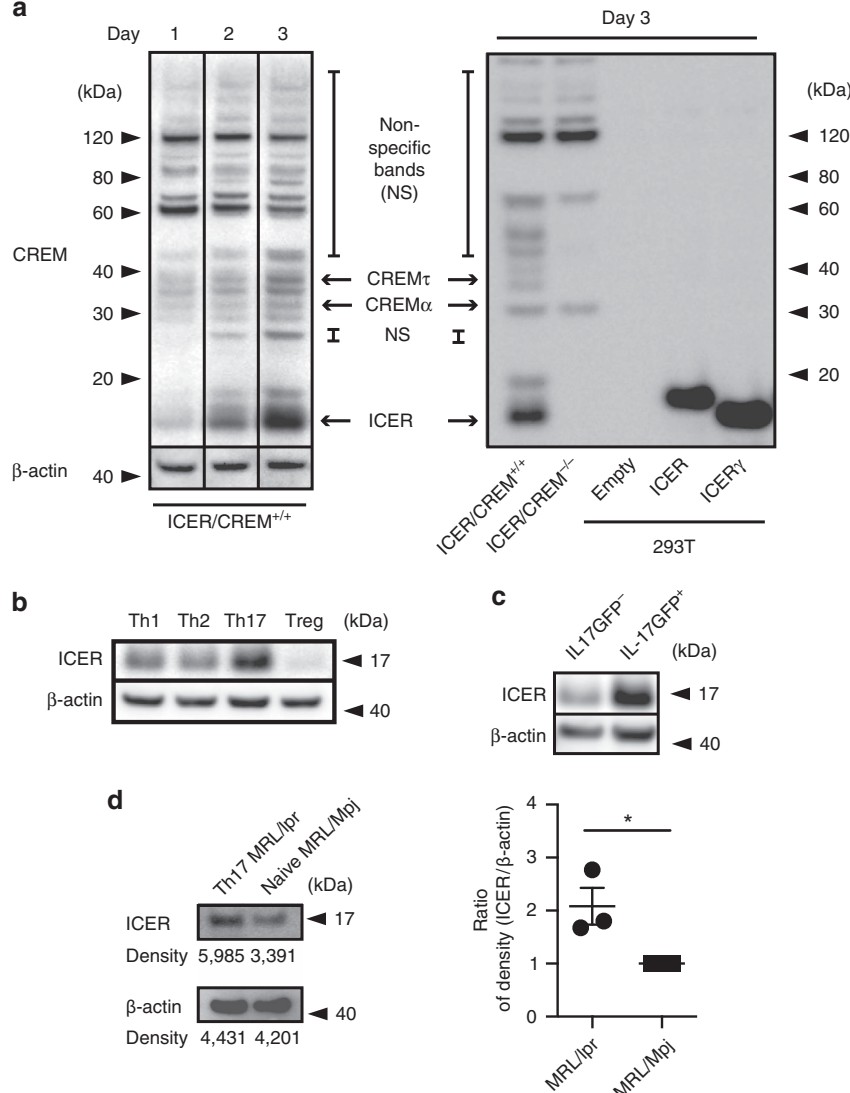

**Figure 1 | ICER is expressed in IL-17-producing murine T cells.** (**a**, left) CREM and β-actin expression in Th17-polarized ICER/CREM$^{+/+}$ murine CD4$^+$ T cells was measured by western blotting at the indicated time points. (**a**, right) CREM expression on day 3 in Th17-polarized ICER/CREM$^{+/+}$ murine CD4$^+$ T cells (right, far left), Th17-polarized B6.ICER/CREM$^{-/-}$ mice CD4$^+$ T cells (right, second left), empty plasmid transfected HEK-293T cells (right, third left), ICER-overexpressing plasmid-transfected HEK-293T cells (right, third right) or ICERγ-overexpressing plasmid-transfected HEK-293Tcells (right, second right) was measured by western blotting. Data are representative of four experiments. (**b**) ICER/CREM and β-actin expression on day 3 of Th0-, Th1-, Th17- and Treg-polarized ICER/CREM$^{+/+}$ mice CD4$^+$ T cells was measured by western blotting. Data are representative of three experiments. (**c**) ICER/CREM and β-actin expression of FACS-sorted GFP$^+$ (IL-17A-producing cells) or GFP$^-$ (IL-17A-non-producing cells) in Th17-polarized B6. IL-17A GFP ICER/CREM$^{+/+}$ CD4$^+$ T cells on day 3 were measured by western blotting. Data are representative of three experiments. (**d**) ICER and β-actin expression of FACS-sorted IL-17A-producing cells from 18-week-old MRL/lpr mice or naive CD4 T cells from 18-week-old MRL/Mpj control mice was determined by western blotting. A representative (of three) blot is shown (left) and densitometric readings from three experiments are shown on the right (*$P < 0.05$; mean ± s.e.m., $n = 4$). See Supplementary Fig. 1 for uncropped scans of the western blot.

detect any ICER induction proving the importance of STAT3 signalling in the induction of ICER.

To follow-up on this data, we searched for defined STAT3-binding sites in the *ICER/CREM* gene. Within the 91 kbp sequence (chr18: 3,251,045–3,342,901) spanning − 15 kbp upstream to + 5.2 kb downstream from the *ICER/CREM* locus (MGI: 88495) one putative STAT3-binding sequence (BS1: + 586/ + 594, 3′-TTCCTGGAA-5′ from the ICER transcription start site) was identified by using the match module of gene regulation (http://www.gene-regulation.com). To investigate whether STAT3 regulates ICER expression at the transcriptional level we cloned the ICER promoter into a

luciferase reporter system. This promoter region includes a CRE-binding site in the p2 promoter (− 116/ − 53), two putative CRE-binding sites (+ 573/ + 580; + 639/ + 646) and a putative STAT3-binding site (+ 586/ + 594) in the first intron (Fig. 3d). As shown in Fig. 3e, the cloned promoter region that defines the STAT3-binding site possessed significantly increased transcriptional activity when the cells were cultured under Th17-polarizing conditions compared with Th0-non-polarized conditions. This activity was completely abrogated in cells transfected with a mutated STAT3 site-mutated promoter (Δ + 586) (Fig. 3d,e), indicating that IL-6 induces ICER expression directly through STAT3 signal transduction.

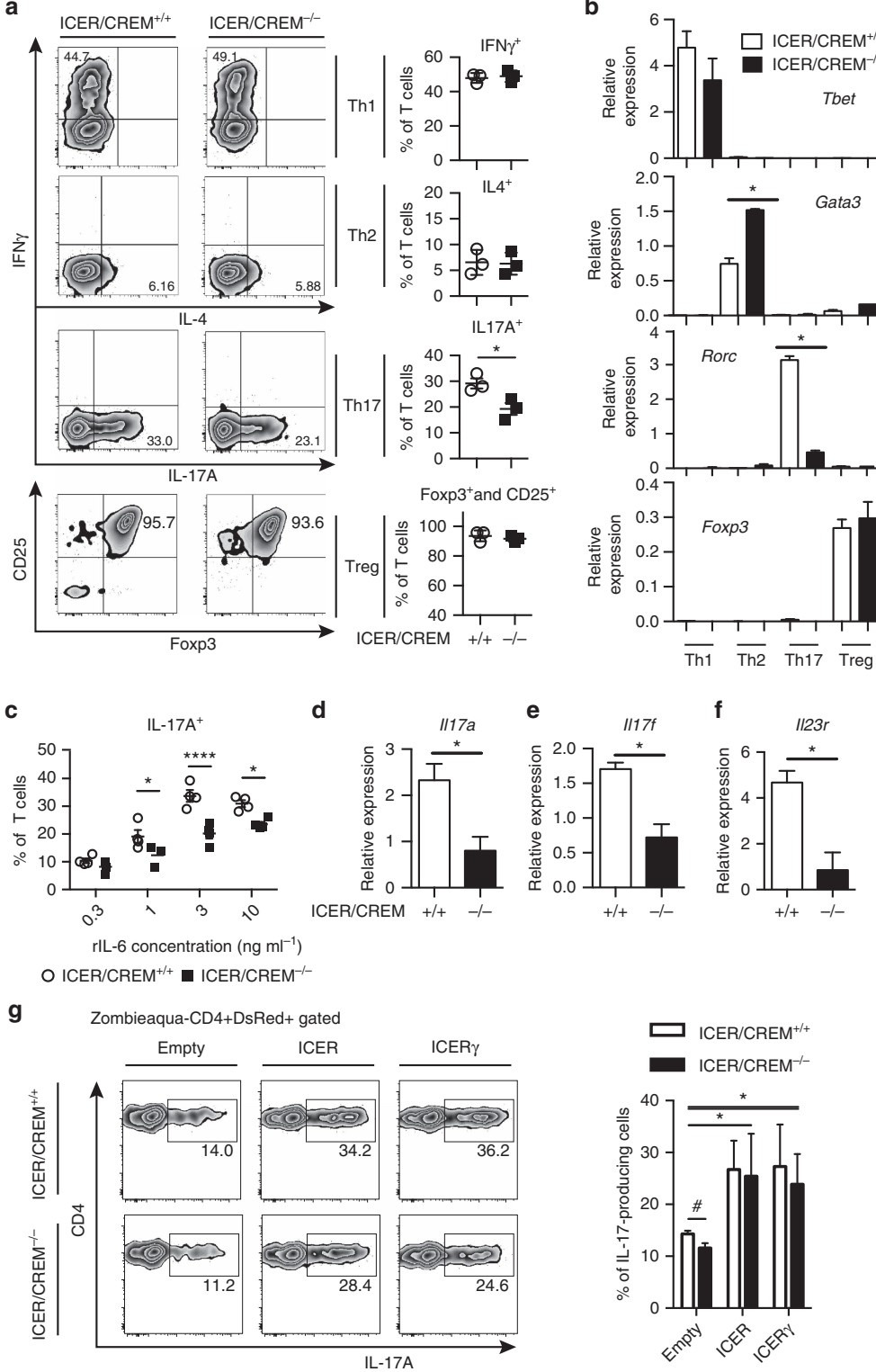

**Figure 2 | Compromised Th17 cell differentiation in ICER/CREM$^{-/-}$ mice.** Naive CD4$^+$ T cells from B6.ICER/CREM$^{+/+}$ mice or B6.ICER/CREM$^{-/-}$ mice were polarized for 3 days as indicated condition. (**a**) Representative flow plots of intracellular expressions of IFNγ or IL-4 and IL-17A (left), the percentages of cells (right) and polarized in indicated condition were measured by flow cytometry (*$P < 0.05$; mean ± s.e.m., $n = 3$). See Supplementary Fig. 2A for FACS gating strategy. (**b**) Real-time PCR analysis of indicated gene expressions in those differentiated cells, relative to β-actin ($n = 3$) (*$P < 0.05$; mean ± s.e.m., $n = 3$). (**c**) The percentage of IL-17-producing cells in Th17-polarized T cells with different concentrations of IL-6. Cumulative results of four independent experiments are shown (****$P < 0.0001$, *$P < 0.05$; mean ± s.e.m., two-way analysis of variance (ANOVA)). (**d–f**) Real-time PCR analysis of (**d**) Il17a, (**e**) Il17f and (**f**) Il23r in Th17-polarized T cells on day 3. A profile representative of three mice is shown (*$P < 0.05$; mean ± s.e.m., $n = 3$). (**g**) Empty vector (Empty), ICER-expressing (ICER) or ICERγ-expressing (ICERγ) plasmids were transfected to Th17-polarized T cells from indicated strains of IL-17A reporter mice on day1, and the percentage of IL-17-producing cells was measured by flow cytometry. A profile representative of three mice is shown on the right (*$P < 0.05$; mean ± s.e.m., two-way ANOVA, Bonferroni, #$P < 0.05$ two-tailed $t$-test, $n = 3$). See Supplementary Fig. 2B for FACS gating strategy.

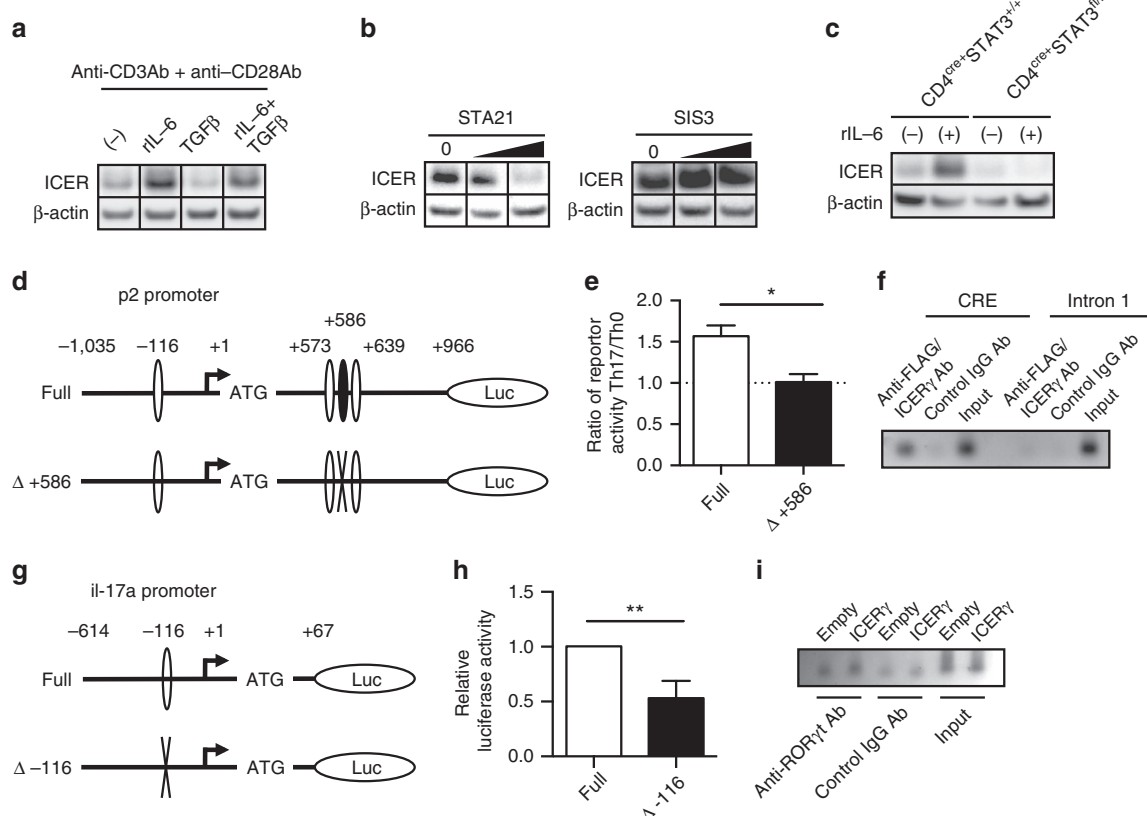

**Figure 3 | ICER is induced by IL-6 via STAT3 and, binds to the IL-17A promoter.** (a–c) CREM and β-actin expression in (a) B6 naive CD4$^+$ T cells cultured in the presence of indicated stimulations on day2, (b) Th17-polarized B6 naive CD4$^+$ T cells cultured in the presence of STAT3 inhibitor (STA21; 0, 10 and 30 μM) or SMAD3 inhibitor (SIS3; 0, 1 and 3 μM) on day3, and (c) Th17-polarized naive CD4$^+$ T cells from indicated strains in the presence or absence of rIL-6 on day3 was measured by western blotting. Data are representative of 3–4 experiments. (d,g) Schematic representations of (d) CRE ($-116/-53$, $+573/+580$ and $+639/+646$) and of Stat3-binding site ($+586/+594$) on the p2 ICER promoter, and (g) CRE ($-116/-109$) on the IL-17a promoter are shown. (e) The full-length ICER promoter region (Full) or a version containing the STAT3-binding site-directed mutation ($\Delta+586$) were transfected to Th17-polarized or Th0-polarized cells. The ratio of reporter activities in those Th17-polarized cells on those Th0-polarized cells was shown (*$P<0.05$; mean ± s.e.m.; $n=3$). (f,i) Binding of (f) FLAG/ICERγ to the CRE or (i) RORγt to the ROR-binding element in Th17-polarized CD4+ T cells from ICER/CREM$^{-/-}$ mice after indicated vector transfection. A comparison to the binding of control Rabbit IgG Ab and sample input. Data are representative of three experiments. For f, the first intron of ICER was used as a negative control for ChIP enrichment. (h) IL-17a promoter activity of Th17-polarized ICER/CREM$^{-/-}$ naive CD4$^+$ T cells was shown. Either full-length mouse IL-17a promoter or a version containing the CRE site-directed mutation ($\Delta$-116) were transfected (**$P<0.01$; mean ± s.e.m.; $n=3$). See Supplementary Fig. 3 for uncropped scans of the western blot.

**ICER enhances RORγt accumulation.** Because previously we had shown that CREMα, which is a suppressive isoform of CREM, bound directly to human *IL-17A* promoter and increased human *IL-17A* promoter activity[14], we considered that ICER could affect similarly *Il17a* promoter activity. To this end we generated FLAG-tagged ICER-overexpressing plasmids (FLAG-ICER and FLAG-ICERγ) and transfected them into ICER/CREM-deficient T cells, which we polarized under Th17 conditions. Transfection of either of the ICER isoforms resulted in increased IL-17 GFP$^+$ T-cell induction compared with those that received empty plasmid (Supplementary Fig. 2). Furthermore, as shown in Fig. 3f, ICER accumulated at the *Il17a* promoter region. The promoter activity of IL-17A decreased after disruption of the CRE-binding site ($-116/-109$) in the *IL-17A* promoter, suggesting the importance of this site in the *il-17A* gene transcription (Fig. 3g,h). Finally, we asked whether ICER binding influenced RORγt accumulation at the ROR-binding element ($-116/-111$). We found that RORγt accumulation at the ROR-binding element was increased when we introduced ICERγ-overexpressing vector compared with empty vector (Fig. 3i). Collectively, in the process of Th17 cell differentiation *in vitro*, ICER is induced by IL-6 though a STAT3 signalling pathway, binds directly to the CRE-binding site of the *IL-17A* promoter and enriches RORγt accumulation at the ROR-binding element.

**ICER/CREM deficiency limits AIGN in mice.** Given that ICER/CREM deficiency had greatly impacted on *in vitro* Th17 cell differentiation, we proceeded to evaluate the relevance of ICER/CREM expression in IL-17-dependent inflammatory conditions. Because the development of kidney damage in AIGN depends on Th17 cells, and IL-17 determines the severity of this induced disease[13,15], we induced AIGN disease in 9- to 11-week-old male ICER/CREM-deficient and -sufficient mice. As shown in Fig. 4a,b, histological sections from ICER/CREM-deficient mice showed less glomerulonephritis, tubulointerstitial inflammation, vasculitis and less protein cast formation compared with those in ICER/CREM-sufficient mice. Accordingly, a significant reduction in proteinuria was observed in ICER/CREM-deficient mice compared with wild-type counterparts (Fig. 4c). Furthermore, when we examined cell infiltration in the diseased kidney, ICER/

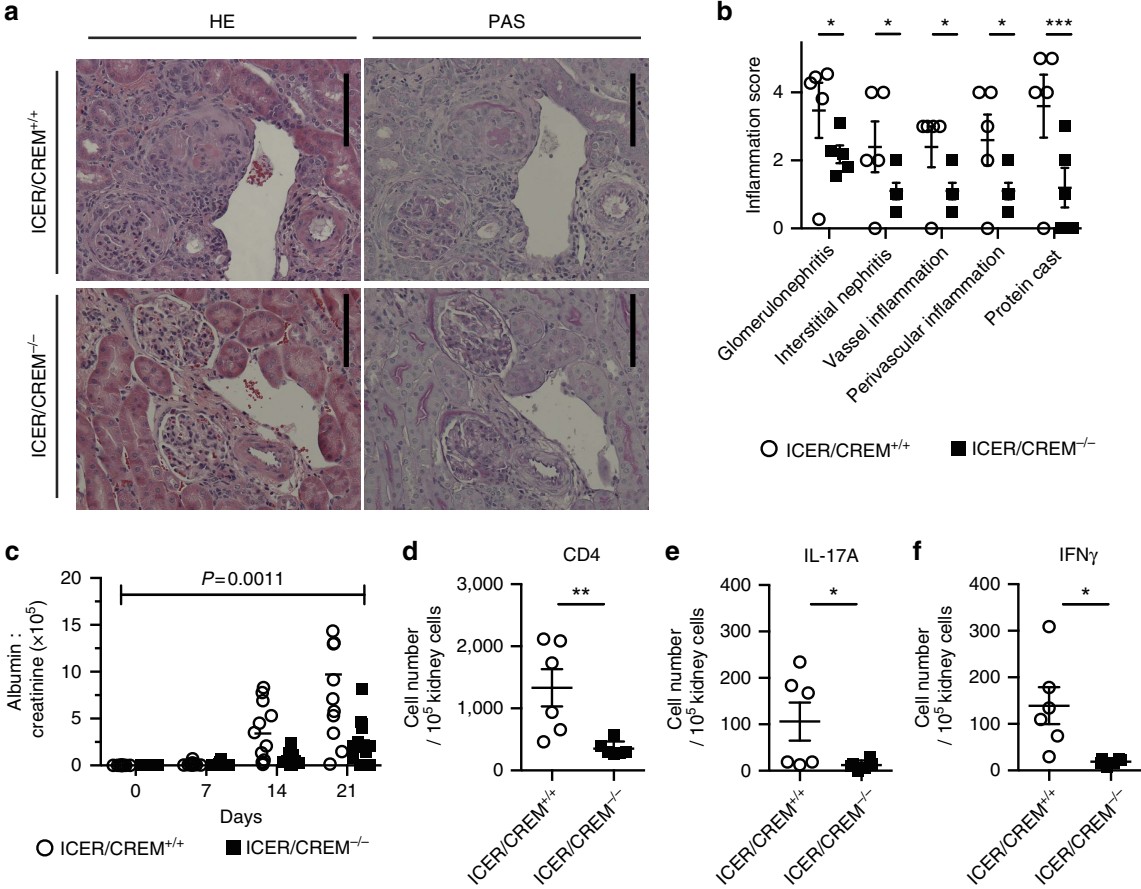

**Figure 4 | ICER/CREM$^{-/-}$ mice are resistant to AIGN.** Mice were immunized with rabbit IgG in complete Freund's adjuvant (CFA) (day $-3$) and then injected intravenously with 200 μl of rabbit nephrotoxic serum (day 0). (**a**) Representative images of haematoxylin and eosin (left) and Periodic acid–Schiff (PAS) (right)-stained sections of kidneys collected by day 21 are shown. Scale bar, 50 μm. (**b**) Inflammation scores of kidney are shown ($*P < 0.05$, $***P < 0.001$; mean ± s.e.m.; two-way analysis of variance (ANOVA); $n = 4$). (**c**) Glomerular injury was quantified by proteinuria (urinary albumin normalized to creatinine) at the indicated time points ($P = 0.0011$; mean ± s.e.m.; two-way repeated measures ANOVA; $n = 12$). (**d–f**) Absolute cell numbers of (**d**) infiltrated CD4$^+$ T cells, (**e**) IL-17A-producing CD4$^+$ T cells and (**f**) IFNγ-producing CD4$^+$ T cells in kidneys collected at day 21 were evaluated by flow cytometry ($*P < 0.05$, $**P < 0.01$; mean ± s.e.m.; $n = 6$).

CREM-deficient mice had fewer infiltrating CD4$^+$ T (CD45$^+$CD90.2$^+$CD4$^+$ cells; Fig. 4d), IL-17A- (CD45$^+$CD90.2$^+$CD4$^+$IL-17A$^+$ cells; Fig. 4e) and interferon gamma (IFNγ)-(CD45$^+$CD90.2$^+$CD4$^+$IFNγ$^+$ cells; Fig. 4f) producing cells than ICER/CREM-sufficient mice. Collectively, ICER/CREM deficiency protects against disease progression in a model of AIGN.

**ICER/CREM deficiency ameliorates EAE.** To further document the role of ICER/CREM in *in vivo* IL-17-dependent pathology, we induced EAE in 9- to 11-week-old ICER/CREM-sufficient and -deficient mice by immunizing them with myelin oligodendrocyte glycoprotein (MOG$_{35-55}$). Disease progression in ICER/CREM-deficient mice was remarkably reduced including lower clinical scores and less body weight changes in ICER/CREM-deficient mice compared with wild-type littermates (Fig. 5b,c). Histological sections of spinal cords showed significantly decreased cell infil-tration and demyelization in ICER/CREM-deficient mice (Fig. 5a). The reduction of infiltrating cells was confirmed by flow cytometry analysis of T cells extracted from spinal cords of diseased mice. Since the number of infiltrated T cells was reduced in spinal cord of ICER/CREM-deficient mice, the absolute numbers of CD4$^+$ (CD45$^+$CD90.2$^+$CD4$^+$; Fig. 5d), IL-17A-

(CD45$^+$CD90.2$^+$CD4$^+$IL-17A$^+$; Fig. 5e) and IFNγ-producing cells (CD45$^+$CD90.2$^+$CD4$^+$IFNγ$^+$; Fig. 5f) were reduced in the spinal cords from ICER/CREM-deficient mice as compared with those from the spinal cords from ICER/CREM-sufficient counterparts, which reflects the decreased disease severity of ICER/CREM-deficient mice. To confirm that this difference is IL-17-related, we immunized ICER/CREM-sufficient and -defi-cient mice with MOG$_{35-55}$ and draining lymph nodes were extracted on day 8. Isolated cells from the lymph nodes were further cultured in *ex vivo* with MOG for 3 days. IL-17A/F and IFNγ concentration were measured by enzyme-linked immuno-sorbent assay. As shown in Fig. 4, IL-17A/F production was decreased in ICER/CREM-deficient mice, whereas IFNγ production was not significantly different (Fig. 5g,h). This further points to the importance of ICER/CREM in the generation of Th17 cells and the production of IL-17.

**ICER/CREM deficiency abrogates lupus disease in B6.*lpr* mice.** CREMα, another repressor isoform of CREM, when over-expressed in lupus-prone mice leads to increased lupus-like autoimmune disease in a Th17-related manner[7,16]. Since ICER/CREM deficiency affected Th17 differentiation *in vitro* and in two different Th17-related disease models, we considered that ICER/

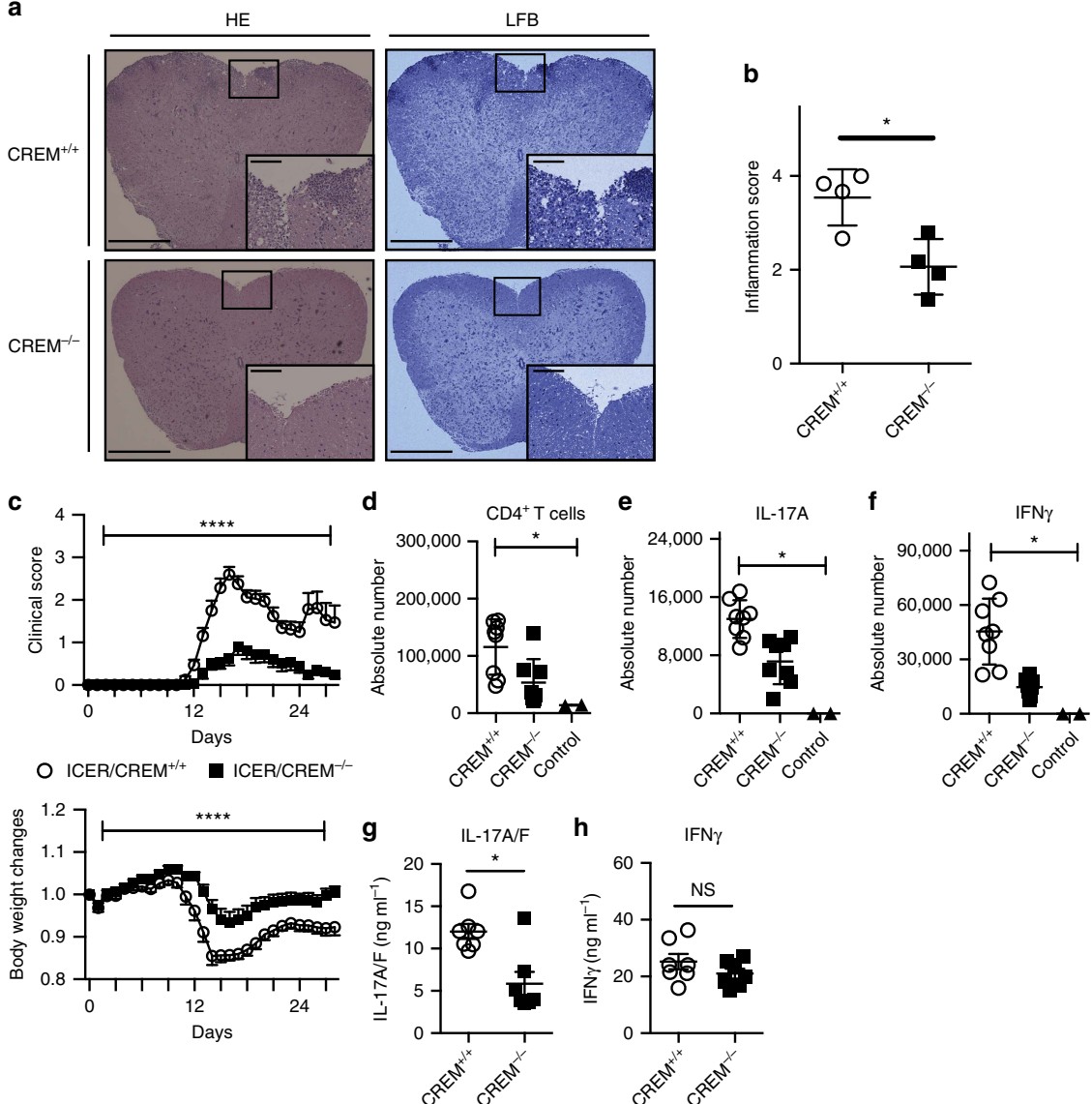

**Figure 5 | ICER/CREM$^{-/-}$ mice are resistant to EAE.** EAE was induced in ICER/CREM$^{+/+}$ and ICER/CREM$^{-/-}$ mice by immunization with MOG$_{35-55}$ emulsified in CFA. (**a**) Spinal cord collected at day 14 from indicated mice were stained with haematoxylin and eosin (left) and luxol fast blue (right) to assess inflammation and myelin content, respectively. Scale bars, 500 or 100 μm (magnified panels). (**b**) Quantitative cumulative data are shown (*$P < 0.05$; mean ± s.e.m.; $n = 4$). (**c**) The clinical scores and %body-weight changes were monitored (****$P < 0.0001$; mean ± s.e.m.; two-way analysis of variance (ANOVA); cumulative results of three independent experiments, $n = 5$, 5 and 6). (**d–f**) Absolute cell numbers of spinal cord-infiltrated (**d**) CD4$^+$ T cells, (**e**) IL-17A-producing CD4$^+$ T cells and (**f**) IFNγ-producing CD4$^+$ T cells from indicated mice were evaluated by flow cytometry ($n = 8$ per group). ICER/CREM$^{+/+}$ mice without inducing EAE were used as a control ($n = 2$) (*$P < 0.05$; mean ± s.e.m.; one-way ANOVA). (**g,h**) Mononuclear cells were collected at day 8 from inguinal lymph nodes and further cultured *ex vivo* with MOG for 3 days. (**g**) IL-17A/F and (**h**) IFNγ concentrations were measured by ELISA (*$P < 0.05$; mean ± s.e.m.; $n = 7$).

CREM deficiency would delay or protect lupus-prone mice form developing disease. We examined the effect of genetic ICER/CREM depletion on the development of spontaneous lupus by crossing ICER/CREM-deficient mice with lupus-prone B6.MRL-Fas$^{lpr}$/J (B6.lpr) mice (both in the C57BL/6J background). We studied female ICER/CREM-deficient B6.lpr mice at 28 weeks of age and found that they displayed smaller spleens and cervical lymph nodes compared with ICER/CREM-sufficient B6.lpr mice (Fig. 6a,b). Immunostaining analysis revealed less deposition of complement 3 (C3) in the glomeruli of ICER/CREM-deficient B6.lpr compared with that in ICER/CREM-sufficient B6.lpr mice (Fig. 6c). Serum levels of anti-dsDNA antibody, proteinuria and double-negative T cells (CD3$^+$CD4$^-$CD8$^-$ cells), and serum

IL-17 were also significantly reduced in the ICER/CREM-deficient mice (Fig. 6d–g). When we examined cell infiltration in the kidneys, ICER/CREM-deficient mice had fewer CD45$^+$-infiltrating lymphocytes (CD45$^+$ per 10$^5$ events), T cells (CD45$^+$CD90.2$^+$) and IL-17A-producing cells (CD45$^+$CD90.2$^+$IL-17A$^+$) than ICER/CREM-sufficient mice (Fig. 6h). Furthermore, as shown in Fig. 6i, the survival curve indicated a significantly prolonged lifespan in B6.lpr.ICER/CREM-sufficient mice (median survival day; 279.5 days for ICER/CREM-sufficient mice and 361.0 days for ICER/CREM-deficient mice; $P = 0.0044$, Gehan–Breslow–Wilcoxon test). Collectively, ICER/CREM deficiency rescues multiple lupus characteristics and extends the lifespan of B6.lpr mice.

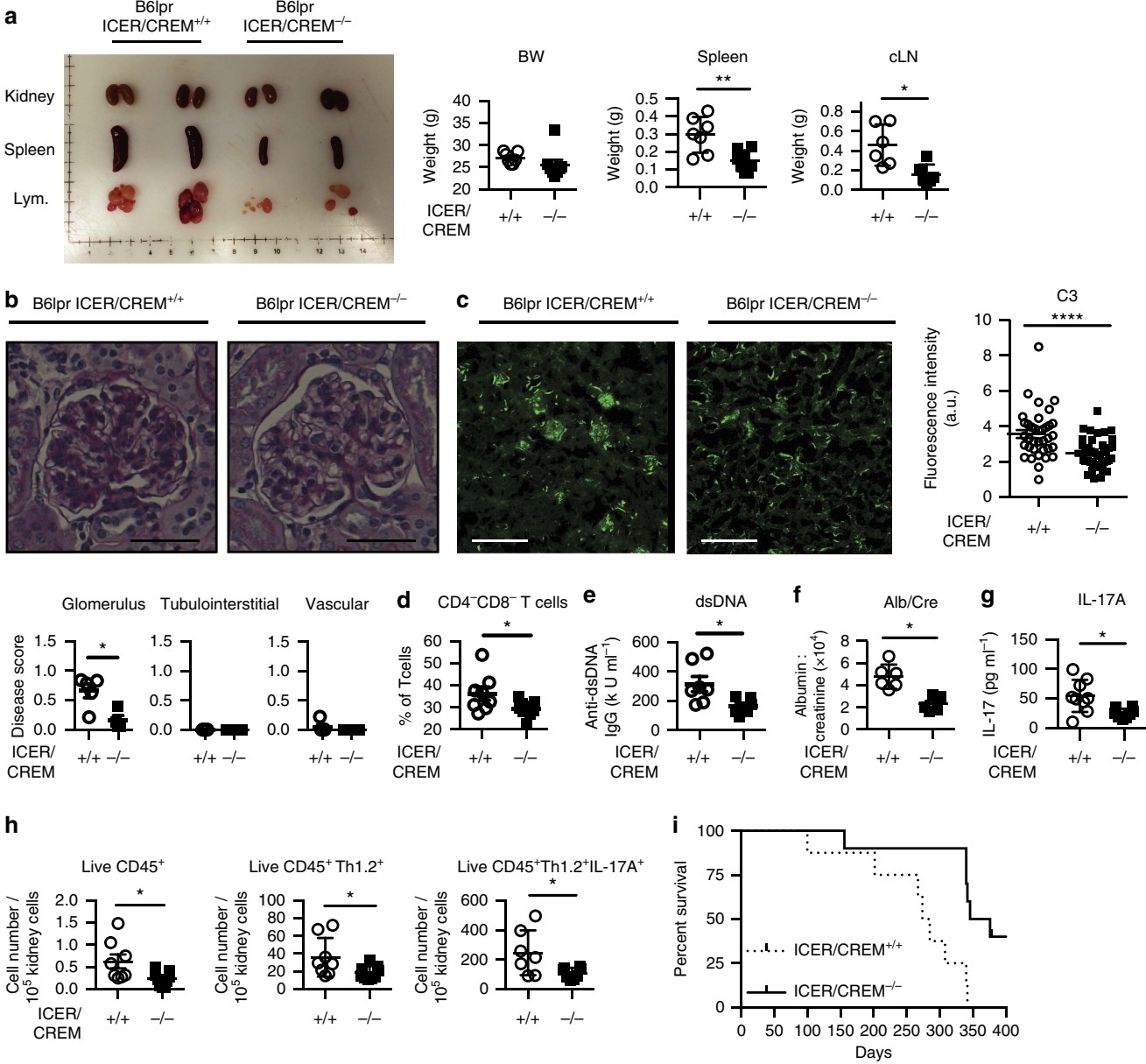

**Figure 6 | B6.*lpr*.ICER/CREM$^{-/-}$ mice display less autoimmunity.** (**a**) B6.*lpr*.ICER/CREM$^{+/+}$ ($n = 7$) and B6.*lpr*.ICER/CREM$^{-/-}$ ($n = 8$) mice were killed at 28 weeks of age. Kidney, spleen ($n = 7$ and 8 mice, respectively) and cervical lymph nodes ($n = 6$ per group) were dissected and weighted (*$P < 0.05$, **$P < 0.01$; mean ± s.e.m.). (**b**) Periodic acid–Schiff (PAS)-stained kidney section and disease scores were shown (*$P < 0.05$; mean ± s.e.m.; $n = 5$ per group). Scale bar, 50 μm. (**c**) C3 deposition in the kidney sections was assessed by direct immunofluorescence. Scale bar, 300 μm. Cumulative data are shown on the right (****$P < 0.0001$; mean ± s.e.m.; cumulative results of total 36 glomerulus from 5 B6.*lpr*.ICER/CREM$^{+/+}$ and total 40 glomerulus from 5 B6.*lpr*.ICER/CREM$^{-/-}$ mice were assessed). (**d**) Percentage of double-negative T cells was evaluated by flow cytometry (8 B6.*lpr*.ICER/CREM$^{+/+}$ and 9 B6.*lpr*.ICER/CREM$^{-/-}$). Serum anti-dsDNA IgG (7 B6.*lpr*.ICER/CREM$^{+/+}$ and 6 B6.*lpr*.ICER/CREM$^{-/-}$) (**e**), urinary albumin to creatinine ratio ($n = 6$ per group) (**f**) and serum IL-17A (7 B6.*lpr*.ICER/CREM$^{+/+}$ and 8 B6.*lpr*.ICER/CREM$^{-/-}$) (**g**) were quantified by ELISA (*$P < 0.05$; mean ± s.e.m.). (**h**) Kidney-infiltrated CD45$^+$ cells, T cells and IL-17A-producing T cells were evaluated by flow cytometry (*$P < 0.05$; mean ± s.e.m.; 8 B6.*lpr*.ICER/CREM$^{+/+}$ and 9 B6.*lpr*.ICER/CREM$^{-/-}$). (**i**) Survival of indicated mice was monitored (8 B6.*lpr*.ICER/CREM$^{+/+}$ and 10 B6.*lpr*.ICER/CREM$^{-/-}$).

**ICER is increased in human Th17 cells and SLE T cells**. Since we found that ICER controls Th17-related experimental inflammation and affects lupus-like manifestations of lupus-prone mice, we sought evidence that it impacts inflammatory mechanisms in human autoimmunity. We sorted peripheral blood mononuclear cells from healthy donors into memory Th1 cells, memory Th2 cells, memory Th17 cells and Treg cells by flow cytometry and compared ICER expression in each population (Fig. 7a). We did not detect ICER expression in any of the T-cell subsets before stimulation (Supplementary Fig. 4). However, after 12 h of activation with anti-CD3 and anti-CD28 Ab, ICER was present at statistically significant amounts only in Th17 cells and not in any of the other three subsets (Fig. 7b). Finally, we stimulated CD4$^+$ T cells with anti-CD3 and anti-CD28 Ab for 4 days and determined the levels of ICER expression by western blotting. As seen in Fig. 7c, SLE T cells express more ICER than normal T cells. Collectively these data suggest that ICER is induced not only in murine but also in human Th17 cells and that its levels are

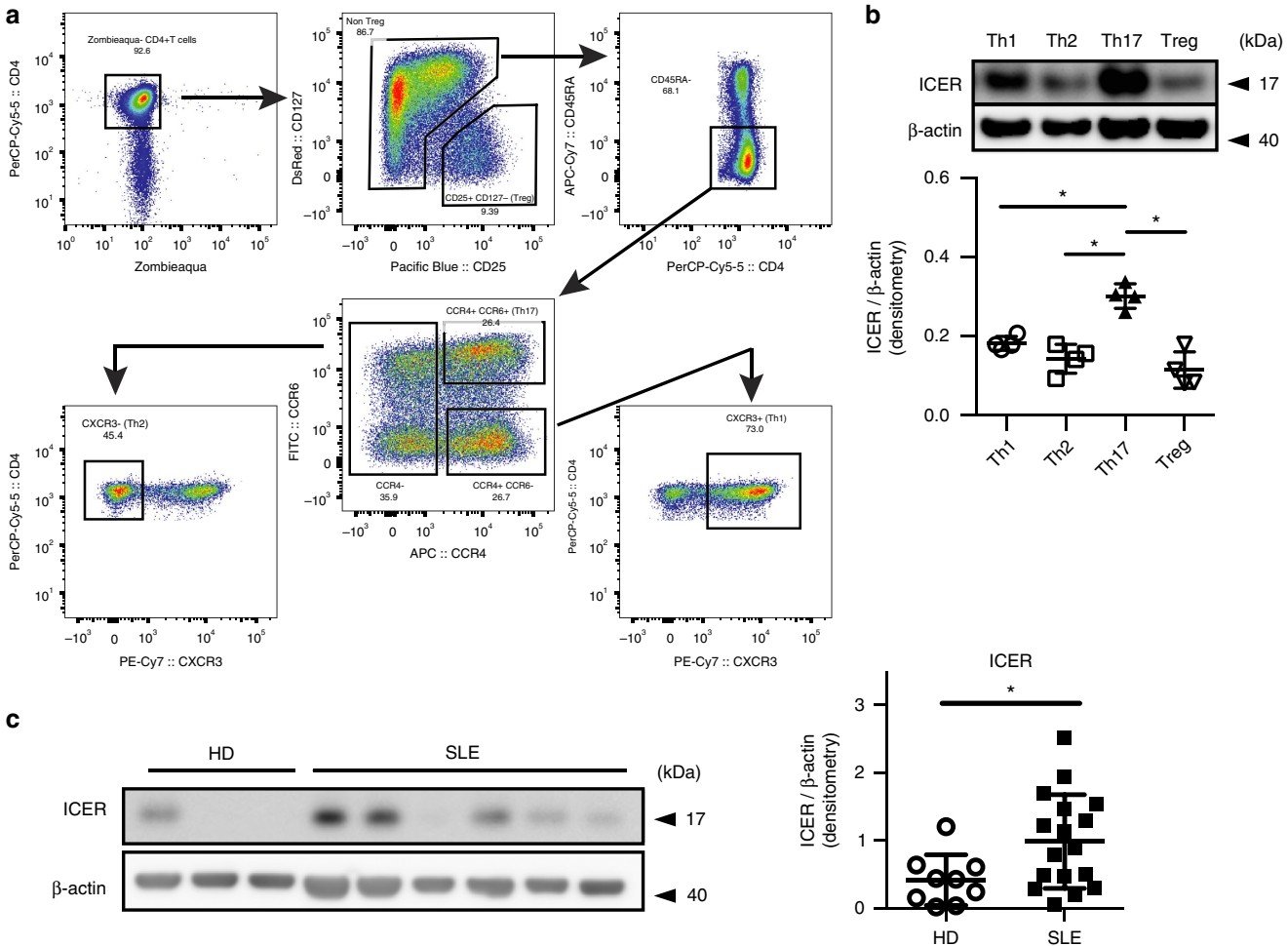

**Figure 7 | Human Th17 cells and SLE T cells express increased amounts of ICER.** (**a**) The gating strategy used to define and sort the primary lymphocytes memory subsets: Th1 (CD4$^+$CD45RA$^-$CXCR3$^+$); Th2 (CD4$^+$CD45RA$^-$CCR6$^-$CCR4$^+$CXCR3$^-$); Th17 (CD4$^+$CD45RA$^-$CCR6$^+$CCR4$^+$); and Treg (CD4$^+$CD25$^+$CD127$^-$). (**b**) FACS-sorted Th1, Th2, Th17 and Treg cells were stimulated with plate-bound anti-CD3 and anti-CD28 antibodies for 12 h. ICER and β-actin expression was determined by western blot (*$P < 0.05$; mean ± s.e.m., $n = 4$). (**c**) Purified CD4$^+$ T cells from healthy donors (HD) and SLE patients were stimulated with plate-bound anti-CD3 and anti-CD28 antibodies. On day 4, cells were collected and CREM subsets and β-actin expression were examined by western blot. The graphs show cumulative data of densitometry for ICER. (*$P < 0.05$; mean ± s.e.m.; HD $n = 9$ and SLE $n = 17$). See Supplementary Fig. 5 for uncropped scans of the western blot and Supplementary Table 1 for patient information.

increased in T cells from SLE patients, suggesting a role of ICER in human disease.

## Discussion

Although the canonical second messenger cAMP was first described as a potent negative regulator of T-cell immune function[17], recent reports have shown conflicting effects of cAMP-related transcription factors on signal transduction and T-cell differentiation. For example, CREB, a well-characterized cAMP-responding transcription factor, has been proposed as a Th17-promoting factor[18] while other studies have emphasized its ability to enhance regulatory T-cell differentiation and Foxp3 transcription[19,20]. CREM, another important cAMP-related transcription factor, is tightly regulated at the epigenetic and post-transcriptional levels. Alternative splice variants of the primary CREM gene generate isoforms that exert opposing effects on target gene expression compared with the full-length proteins[17]. It appears though that the relative abundance of CREM splice variants in different tissues and organs accounts for specificity of the control of target gene expression. Understanding the expression and the function of the different CREM isoform(s) during the development of an immune response is thus warranted.

Here we demonstrate a novel important role of ICER in the generation of Th17 cells and in the development of IL-17-related diseases in mice, including AIGN, EAE and SLE. Although previous reports suggested that ICER exerts a suppressive function in effector T cells[10,11], we provide evidence demonstrating that ICER is the most prevalent CREM splice variant expressed during Th17 differentiation and is preferentially expressed during Th17 generation compared with other T helper subsets. We show that ICER is involved in Th17 T-cell generation and is implicated in Th17/IL-17-dependent autoimmune diseases.

Th17 differentiation requires the presence of IL-6 and its transducing signalling partner STAT3. Our findings provide evidence showing that ICER is a downstream partner of the IL-6/STAT3 signalling pathway. Moreover, induction of ICER during Th17 differentiation is independent of TGFβ. This suggests that ICER may be involved in the induction of pathogenic Th17 T cells, which are generated in the absence of TGFβ (ref. 21).

IL-17 is important for the development of human and murine inflammatory pathology[4,22–24]. Here we present strong evidence that ICER/CREM is required for the expression of both organ-specific and systemic autoimmunity. Lack of ICER/CREM mitigates the development of AIGN as measured histologically and clinically. Importantly, the number of IL-17-producing cells in the kidney of ICER/CREM knockout mice is significantly diminished. Similarly, lack of ICER/CREM decreases EAE clinical and histopathologic severity, as well as the frequency of peripheral IL-17-producing cells. Of interest, T cells from draining lymph nodes of ICER/CREM-deficient mice display normal IFNγ production, but a limited production of IL-17, pointing out a predominant effect of ICER/CREM on Th17 differentiation. In the absence of ICER/CREM, B6.lpr lupus-prone mice display less systemic autoimmunity (anti-dsDNA antibody) and renal pathology. This phenotype is linked to decreased serum levels of IL-17 levels and to reduced numbers of IL-17-producing cells in the kidney. These results underscore the importance of ICER/CREM in the production of IL-17 and autoimmune disease pathogenesis[25–27].

In humans we show that Th17 CD4+ T-cell-differentiated subset express higher level of ICER than Th1, Th2 and Treg subsets. More importantly, we observed increased expression of ICER in CD4+ T cells isolated from SLE patients, underscoring the translational value of our finding. SLE patients display increased serum levels of IL-17A, expanded frequency of IL-17-producing T cells in the peripheral blood and a massive infiltration of Th17 cells in organs involved by the disease, like the kidneys[28,29]. Furthermore, IL-17A levels correlate with SLE disease activity. Sera from patients with SLE show increased levels of IL-6 (ref. 30), and STAT3 (ref. 31) is elevated in SLE T cells, abnormalities that could account for the augmented expression of ICER.

Robust activation of the cAMP-related signalling pathway inhibits T-cell activation[17] while precise control of the ratio of cAMP-related transcription factors favours a differentiation towards a Th17 phenotype[18]. Since ICER negatively regulates cAMP-related transcription, ICER may play a major role in this fine-tuning process. Since it has been reported that levels of cAMP are increased in ICER/CREM-deficient mice[32], and cAMP is known to regulate glycolysis and glycolysis plays a critical role in Th17 cell differentiation, we speculate that ICER/CREM affects the glycolysis pathway. In SLE, both CREMα overexpression and induced ICER act in concert to promote a pro-inflammatory Th17 phenotype, by allowing increased RORγt accumulation.

In conclusion, we have identified ICER as a novel requisite determinant of Th17 cell generation. It acts downstream of STAT3, and after its binding to the IL-17 promoter, it enhances the accumulation of the canonical Il17 transcription factor RORγt. Importantly, genetic elimination of all ICER/CREM isoforms suppressed the development of organ-specific and systemic autoimmunity. The translational importance of our work is highlighted by our findings showing that ICER expression is increased in SLE T cells, as well as in human Th17 memory cells. Our findings identify a unique role for ICER, which impacts both organ-specific and systemic autoimmunity in a Th17-dependent manner.

## Methods

**Human samples and cell lines.** Patients who fulfilled the criteria for the diagnosis of SLE as set forth by the American College of Rheumatology[33] and healthy individuals were enrolled. The BIDMC Institutional Review Board approved the study protocol (2006-P-0298). Informed consent was obtained from all study subjects. The disease activity for each patient was calculated using the clinico-laboratory index SLE Disease Activity Index [34]. Age-, sex- and ethnicity-matched healthy individuals were chosen as controls (Supplementary Table 1). Peripheral venous blood was collected in heparin-lithium tubes, and total human T cells were

purified as described previously[35]. In short, T cells were isolated by negative selection (RosetteSep, Stem Cell Technologies) before density gradient purification (Lymphoprep, Nycomed). HEK-293T cells were purchased from American Type Culture Collection (ATCC; Manassas, VA) and has been tested for mycoplasma by ATCC.

**Mice.** SV129/Bl6.ICER/CREM$^{-/-}$ mice were originally cloned by Guenther Schuetz, DKFZ Heidelberg[36]. Animals were crossed to C57BL/6J mice for over nine generations to transfer the ICER/CREM$^{-/-}$ locus to the B6 background. Female B6.MRL-Faslpr/J (B6.lpr), C57BL/6-Il17atm1Bcgen/J (IL-17GFP), STOCK Tg (Cd4-cre)1Cwi/BfluJ (CD4$^{cre+}$), MRL/MpJ-Fas$^{lpr}$/J and B6.129S1-Stat3tm1Xyfu/J (STAT3$^{fl/fl}$) mice were purchased from The Jackson Laboratory. B6.lpr.ICER/CREM$^{-/-}$ mice were made by crossing B6.ICER/CREM$^{-/-}$ mice with B6.lpr mice. CD4$^{cre+}$. STAT3$^{fl/fl}$ mice were made by crossing STAT3$^{fl/fl}$ mice with CD4$^{cre+}$ mice. B6.ICER/CREM$^{-/-}$.IL-17GFP mice were made by crossing B6.ICER/CREM$^{-/-}$ mice with IL-17GFP mice. Animals were killed at the end of their 8–12 weeks of life for in vitro culture experiments and indicated week for in vivo experiments. All mice were maintained in an SPF animal facility (Beth Israel Deaconess Medical Center). Experiments were approved by the Institutional Animal Care and Use Committee of BIDMC. All mice were genotyped to validate claimed strain.

**Single-cell isolation.** Spleens and lymph nodes were excised and single-cell suspensions were obtained. Kidneys were perfused with PBS and digested with collagenase type IV (100 μg ml$^{-1}$) (Worthington Biochemical) in Hank's balanced salt solution (HBSS) for 30 min (37 °C). Infiltrating lymphocytes in spinal cords were isolated as previously described[37]. Briefly, spinal cords were digested with collagenase type IV (100 μg ml$^{-1}$) in HBSS for 20 min (37 °C). Cell suspensions from digested spinal cords were subjected to density separation using Optiprep density gradient medium (Sigma-Aldrich) followed by flow cytometry.

***In vitro*** **T-cell differentiation.** Naive CD4$^+$ T cells were purified by mouse CD4$^+$CD62L$^+$ T Cell Isolation Kit II (Miltenyi Biotec). Purified naive T cells were stimulated with plate-bound goat anti-hamster antibodies, soluble anti-CD3 (0.25 μg ml$^{-1}$,145-2C11; Biolegend) and anti-CD28 (0.5 μg ml$^{-1}$, 37.51; Biolegend) for Th0-non-polarized condition culture. In addition to Th0-non-polarized condition, following stimulation was used for each polarized condition: IL-12 (20 ng ml$^{-1}$; R&D Systems) and anti-IL-4 (10 μg ml$^{-1}$, C17.8; Biolegend) for Th1; IL-4 (100 ng ml$^{-1}$; R&D Systems), anti-IL-12 (10 μg ml$^{-1}$; Biolegend) and anti-IFNγ (10 μg ml$^{-1}$, XMG1.2; Biolegend) for Th2; IL-6 (3 ng ml$^{-1}$ or indicated concentration; R&D Systems), TGF-β1 (0.3 ng ml$^{-1}$; R&D Systems), anti-IL-4 (10 μg ml$^{-1}$, C17.8; Biolegend) and anti-IFNγ (10 μg ml$^{-1}$; XMG1.2; Biolegend) for Th17; and IL-2 (20 ng ml$^{-1}$; R&D Systems), TGF-β1 (3 ng ml$^{-1}$; R&D Systems), anti-IL-4 (10 μg ml$^{-1}$, C17.8; Biolegend) and anti-IFNγ (10 μg ml$^{-1}$, XMG1.2; Biolegend) for Tregs. For signal transduction studies, STA21 (STAT3 inhibitor; Santa Cruz Biotechnology) and SMAD3 inhibitor (SIS3; Santa Cruz Biotechnology) were added to cultures on day 0.

**Western blotting.** Cell lysate was separated on NuPAGE 4–12% Bis-Tris Gel (Life Technologies) and proteins were transferred to a nitrocellulose membrane. Following antibodies were used: anti-CREM1 Ab (clone X-12, Santa Cruz, 1:500); Stat3 Mouse mAb (clone 124H6, Cell Signaling, 1:500); Phospho-Stat3 (Tyr705) Mouse mAb (clone 3E2, Cell Signaling, 1:500); anti-β-actin (clone AC-74, Sigma-Aldrich, 1:10,000); goat anti-mouse IgG coupled with horseradish peroxidase (catalogue# sc-2005, Santa Cruz, 1:3,000); and goat anti-rabbit IgG coupled with horseradish peroxidase (catalogue# sc-2004, Santa Cruz, 1:3,000). The ECL system (Amersham) was used for detection.

**Flow cytometry.** Following antibodies were used for flow cytometry analysis: for mouse, CD4 (clone GK1.5); CD8a (clone 53-6.7, 1:100); CD19 (clone 605, 1:100); CD25 (clone PC61, 1:100); CD44 (clone IM7, 1:100); CD45 (clone 30-F11, 1:100); CD62L (clone MEL-14, 1:100); CD90.2 (clone 53-2.1, 1:100); IL-17A (clone JC11-18H10.1, 1:50); IFNγ (clone XMG1.2, 1:50); and IL-4 (clone 11B11, 1:50) were purchased from BioLegend. CD3α (clone 17A2, 1:100) and Foxp3 (clone FJK-16s, 1:50) were purchased from eBioscience. For human, CD45RA (clone HI100, 1:100), CD25 (clone BC96, 1:100), CD127 (clone A019D5, 1:100), CCR4 (clone L291H4, 1:100), CCR6 (clone G034E3, 1:100) and CXCR3 (clone G025H7, 1:100) were purchased from BioLegend. A CD4 (clone SK3, 1:100) was purchased from eBioscience. A 7AAD (surface) or a Zombie Aqua Fixable Viability Kit (intracellular) staining was performed for eliminating dead cells (BioLegend). Surface staining was performed on ice for 20–30 min. Absolute cell numbers were calculated on the basis of the percentage of each cell population. For intracellular staining, collected cells were stimulated for 4 h in culture medium with phorbol myristate acetate (500 ng ml$^{-1}$; Sigma-Aldrich), ionomycin (1.4 μg ml$^{-1}$; Sigma-Aldrich) and monensin (1 μl ml$^{-1}$; BD Biosciences), except Foxp3 detection. Cytofix/Cytoperm and Perm/Wash buffer (BD Biosciences; for IL-17A/IL-4/IFNγ staining) or Mouse Regulatory T-cell staining kit (eBioscience; for Foxp3 staining) were used for fixation and permeabilization. All flow cytometry data were acquired

on a BD LSRII and analysed with FlowJo. All procedures were performed according to the manufacturer's instructions.

**Western blotting for formalin-fixed and FACS-sorted cells.** Cells from 18-week-old MRL/lpr mice and MRL/Mpj mice spleen and lymph node were stimulated in culture medium contains phorbol myristate acetate (500 ng ml$^{-1}$; Sigma-Aldrich), ionomycin (1.4 µg ml$^{-1}$; Sigma-Aldrich) and monensin (1 µl ml$^{-1}$; BD Biosciences) for 3 h. Collected cells were stained by a Zombie Aqua Fixable Viability Kit, CD4, CD8, CD90.2, CD19, CD62L antibodies (for surface) and IL-17A antibodies (for intracellular) as described before. After staining, ZA-CD19-CD62L-CD8-Th1.2 + IL-17 + cells (for IL-17-producing cells) and ZA-CD19-CD62L + CD8-CD4 + Th1.2 + IL-17- cells (for naive CD4 T cells) were subsequently sorted by BD FACS Aria II (five lasers 355, 405, 488, 561 and 640 nm; BD Bioscience). Sorted cells were washed with PBS, then lysed in Extraction Buffer EXB Plus (Qproteome FFPE Tissue Kit, Qiagen), which contains 6% β-mercaptoethanol (Sigma-Aldrich), on ice for 5 min, followed by heat denaturation at 100 °C for 20 min. Then samples were incubated at 80 °C for 2 h with vortexing every 2 min, followed by centrifuging them for 15 min in cold room. A volume of 5 µl of NuPAGE LDS Sample Buffer (4 ×) (Invitrogen) was added to the supernatant containing the extracted proteins.

**ELISA.** Following ELISA kits were used: ELISA MAX Deluxe SET Mouse IL-17A/F; IFN-γ and IL-2 (BioLegend); mouse anti-dsDNA IgG ELISA (Alpha Diagnostic Intl. Inc.); Parameter Creatinine Kit (R&D Systems); and Mouse Albumin ELISA Quantitation Set (Bethyl Laboratories). All procedures were performed according to the manufacturer's instructions.

**RNA isolation and quantitative PCR.** Total RNA was prepared using the Qiagen RNeasy Mini kit (Qiagen) and RNA was reverse transcribed into cDNA using the EcoDry Premix (Oligo dT) (Clontech) according to the manufacturer's instructions. Quantitative PCR was performed by using LightCycler 480 SYBR Green I Master (Roche). Target genes were detected using intron-spanning primers. Gene expression was assessed by comparative CT method. Primers information is described in the Supplementary Table 2.

**ICER subset-overexpressing vectors.** Plasmid contains wild-type ICER coding sequence (corresponding to GenBank. AJ311667.1) was kindly gifted by Dr Shogo Endo[38], and subcloned into pIRES2-DsRed-Express vector (Clontech Laboratories Inc.). Following oligonucleotide sequences were used: 5'-TGATCTCGAGCATGG CTGTAACTGGAGATG-3'; and 5'-TGCTGGATCCCGTTACTCTACTTTATGG CAAT-3'. ICERγ overexpression vector and N'-FLAG-tagged overexpressing vectors were generated using Q5 site-directed mutagenesis kit (New England Biolabs). Following oligonucleotide sequences were used: 5'-CTGCCACAGGT GACATGCCAAC-3'; and 5'-CAGTTTCATCTCCAGTTACAGC-3'. For generating N'-FLAG-tagged overexpressing plasmid, FLAG sequence (DYKDDDDK) were inserted N' terminal of ICER-overexpressing plasmid or ICERγ-overexpressing plasmid. Following oligonucleotide sequences were used: 5'-GACGATGACAAGGCTGTAACTGGAGATGAAAC-3'; and 5'-ATCCT TGTAGTCCATCCTGAGATCTGAG-3'. All constructs were validated by DNA sequencing. All procedures were performed according to the manufacturer's instructions.

**Transfection of overexpressing vectors.** For transfection to HEK-293T cells, plasmids were transfected to 60–80% confluent HEK-293T cells by poly-ethylenimine 'Max' (Polysciences, Inc.) according the manufacturer's protocol and cultured 2 days in culture media. For ICER-overexpressing experiments in murine primary T cells, cells were collected 1 day after starting culture and empty vector, ICER-overexpressing plasmid, or ICERγ-overexpressing plasmid were transfected using the Amaxa Mouse T Cell Nucleofector Kit with the X-001 programme (Amaxa) according to the manufacturer's protocol. The supernatant of each culture was saved in 4 °C during transfection and recovery. After 4 h recovery at 37 °C, cells were again cultured in those supernatant for 2 days. The efficacy of the transfection always exceeds 10%.

**Luciferase reporter constructs.** To generate the mouse ICER P2 promoter luciferase promoter construct, we TA-cloned the p2 promoter from genomic DNA of C57BL/6 mouse using KOD XL DNA Polymerase (Thomas Scientific) and pGEM-T easy Vector System (Promega). Following primers were used: Ms_pICER_F, 5'-CACTGTGGAGCCTGGTATGT-3'; and MS_pICER_R, 5'-CCC ACTTGTCACTAGGCAGG-3'. Using TA-cloned mouse ICER P2 promoter, a PCR was performed with primers Mlu1_mpICER_F; 5'-GTTAACGCGTCACTG TGGAGCCTG-3' and Bgl_mpICER_R; 5'-CGATAGATCTCCCACTTGTCA CTAG-3' for adding restriction enzyme sites (Mlu1 and Bgl2 for 5'-end and 3'-end, respectively). Then this amplicon was ligated into pGL3 vector using T4 DNA Ligase (M202; New England BioLabs) to generate ICER_p2_pGL3_vector. Q5 site-directed mutagenesis kit (New England BioLabs) was used for site-directed mutagenesis. Following DNA oligonucleotide was used for site-directed

mutagenesis at the +586/+594 STAT3-binding site (TTCCTGGAA) within ICER_p2_pGL3_ vector; 5'-TCGTGTTTCAGCGGGGAA-3 and 5'-ATGTAA TGACGTCAGCCC-3'. The pGL4 mIL-17A promoter (pGL4 mIL-17 0.6 kb promoter) was a gift from Warren Strober (Addgene plasmid # 20126)[39]. Q5 site-directed mutagenesis kit (New England BioLabs) was used for site-directed mutagenesis. Following DNA oligonucleotide was used for site-directed mutagenesis at the −116/−109 CRE site (TGACCTCA) within pGL4 mIL-17A_vector; 5'-TTTGAGGATGGAATCTTTACTCAAATG-3' and 5'-GCACAGAACCACCCCTTT-3'. All Sequences were verified (Genewiz). All procedures were performed according to the manufacturer's instructions.

**Luciferase assay.** Luciferase reporter plasmid was transfected using the Amaxa Mouse T Cell Nucleofector Kit with the X-001 programme (Amaxa) on day 2 of culture. Each reporter experiment included 200 ng renilla luciferase construct as an internal control. Luciferase activity was quantified using the Promega Dual Luciferase Assay System (Promega) on day 3 of culture according to the manufacturer's instructions.

**Chromatin immunoprecipitation assays.** Freshly isolated naive CD4 + T cells from B6. ICER/CREM$^{-/-}$ mice were cultured in Th17 polarizing condition for 3 days. Plasmids were transfected as described above on day 1. Collected cells were lysed and ChIP assay was performed using MAGnify Chromatin Immunoprecipitation System (Invitrogen). For RORγt immuno-precipitation, Ds-Red + vector-transfected cells were sorted by flow cytometry before ChIP assay. ANTI-FLAG antibody produced in rabbit (Sigma-aldrich), and anti-Human/Mouse ROR gamma (t) Purified AFKJS-9; eBioscience) were used for immunoprecipitation. Following primer pairs were used: 5'-CGTCATAAAGGGGTGGTTCT-3' and 5'-TTACGTCAAGAGTGGGTTGG-3' for CRE/ROR-binding element; and 5'-GAACTGGAAATGAAACCTTTGG-3' and 5'-TTTCATCACAGCAACCCT TC-3' for non-CRE site. All procedures were performed according to the manufacturer's instructions.

**AIGN model.** A volume of 50 µl Freund's Incomplete Adjuvant (Thermo Scientific), 250 µg HR37a *Mycobacterium tuberculosis* powder (Difco) and 0.1 mg ml$^{-1}$ Rabbit IgG (Pierce) were emulsified and injected in each flank subcutaneously in 8-week-old male mice 3 days before starting the experiment. On day 0, 100 µl Rabbit anti-mouse GBM serum (kindly gifted from T. N. Mayadas, Pathology of Harvard medical school, Boston) was injected intravenously. Urine samples were collected on day 0, 7, 14 and 21 of the experiment, and urinary albumin and creatinine were quantified by ELISA as above. On day 21, kidneys were excised from the mice and processed for flow cytometry and histology. Clincial and histologic scores were evaluated in a blind manner.

**EAE disease model.** On day 0, 8-week-old male mice were immunized subcutaneously with 50 µg MOG$_{35-55}$ peptide emulsified in complete Freund's adjuvant (Sigma) containing 4 mg ml$^{-1}$ *M. tuberculosis* extract (H37Ra; Difco) distributed between the two hind flanks. On days 0 and 2, 150 ng per mouse pertussis toxin (List Biological Laboratories) was given by intraperitoneal injection. Mice were monitored and weighted daily until day 28 of the experiment, and clinical scores were given as follows: 1, limp tail; 2, hindlimb paresis; 3, hindlimb paralysis; 4, tetraplegia; and 5, moribund. Clinical and histologic scores were evaluated in a blind manner.

**Priming assay for MOG immunizations.** Mice were immunized with 50 µg MOG$_{35-55}$ peptide emulsified in complete Freund's adjuvant into both flanks. Eight days after the immunization, inguinal lymph nodes were excised.

**Histological staining and analysis.** For EAE models, sections from 10% formalin-fixed spinal cords were stained with haematoxylin and eosin and luxol fast blue. Histology was scored by an investigator blinded to experimental group. Spinal cord sections were scored by an investigator blinded to experimental group as follows: 0, no infiltration (<50 cells); 1, mild infiltration of nerve or nerve sheath (50–100 cells); 2, moderate infiltration (100–150 cells); 3, severe infiltration (150–200 cells); and 4, massive infiltration (>200 cells). For AIGN model and B6.lpr mice, sections from 10% formalin-fixed kidneys were stained with periodic acid-schiff. Renal lesions were evaluated blindly by a nephro-pathologist as described previously[14] for B6.lpr mice or criteria described in http://www.bolderbiopath.com/systemic-lupus-erythematosus-sle/ for AIGN model. Sections of frozen kidneys were stained with Hoechst 33258 (Life technologies; 1:1,000) and Fluorescein-conjugated goat IgG to mouse complement C3 (MP Biomedicals; 1:200). Then specimens were analysed with a Nikon Eclipse Ti confocal microscope. Images were analysed with EZ-C1 v.3.7 and Image J.

**Assessment of ICER expression from isolated human T cells.** Isolated T cells from SLE patients and healthy controls were stimulated with pre-coated 1 µg ml$^{-1}$ anti-CD3 (clone OKT3, BioXcell) and 1 µg ml$^{-1}$ anti-CD28 (clone 28.2, BioLegend) for 4 days.

**Human primary CD4$^+$ lymphocyte subset sorting.** Peripheral blood mononuclear cells were isolated from Trima-collar (Blood Donor Center, Boston Children Hospital) by density gradient centrifugation (Lymphoprep, Nycomed). CD4$^+$ T cells were enriched using CD4$^+$ T-cell isolation kit II (Miltenyi Biotec). After staining, primary lymphocyte subsets were subsequently sorted by BD FACS Aria II (five lasers 355, 405, 488, 561 and 640 nm; BD Bioscience).

**Statistics.** Samples sizes were chosen based on previous experience in our laboratory. Statistical analyses were performed in GraphPad Prism version 6.0 software. Statistical significance was determined by t-tests (two-tailed) for two groups or one-way analysis of variance with Bonferroni's multiple comparisons tests for three or more groups. For AIGN model and EAE model, clinical scores and body weight changes of each treatment group were compared using two-way analysis of variance. P values of <0.05 were considered statistically significant. No data were excluded from the statistical analysis.

**Data availability.** The data that support the findings of this study are available from the corresponding author on request.

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

## Acknowledgements

This work was supported by NIH grant (NIAID #R37AIO49954 to G.C.T., and #R01AR060849 to V.K.), 2011 Japan Sumitomo Life Welfare and Culture Foundation Subsidy for Invitation of Overseas Scientists (to N.Y.), 2013 Japan Society for the promotion of science postdoctoral fellowships for research abroad (to N.Y.) and SICPA Foundation grant (to D.C.). We thank Lucia Novelli and Robin Bosse for technical assistance. We thank Cox Terhorst for reviewing the manuscript critically and providing constructive comments.

## Author contributions

N.Y. designed all research studies, conducted experiments, acquired data, analysed data and wrote the manuscript; G.C.T. designed research studies, conducted experiments and wrote the manuscript; D.C. and M.P.K. acquired and analysed human data; M.M., K.O., T.K. and M.K. acquired *in vivo* murine data (EAE, B6.lpr); F.R. and J.C.C. acquired and analysed data for the AIGN experiment; T.N.M. provided rabbit anti-mouse GBM serum and analysed AIGN data; V.C.K. provided CD4$^{cre+}$ STAT3$^{fl/fl}$ mice and human samples; K.T. provided ICER/CREM-deficient mice; S.J.B. generated vectors and edited the manuscript.

## Additional information

**Competing financial interests:** The authors declare no competing financial interests.

