## [Peer Review File · Nature Communications]

Reviewers' comments:

Reviewer #1

Expert in EAE and IL-17a

(Remarks to the Author):

This study shows that the inducible cAMP early repressor (ICER) is involved in development of Th17 cells that are pathogenic in autoimmune diseases. The study is novel and the data is mostly convincing, especially in animal models with less extensive or convincing human data using PBMC from patients with SLE.

Specific comments

- 1) The data with in vitro polarized Th1, Th2, and Th17 cells is based on IL-6 and TFG-beta polarized Th17 cells from naive mice, activated with anti-CD3 and CD28. It is not clear whether ICER is induced by cell activation in vitro or expressed by Th17 cells in vivo. This would be enhanced with data on ex vivo Th17 from mice with autoimmune diseases either without re-stimulation or following activation with IL-1 and IL-23, the activation stimulus for memory Th17 cells.
- 2) The in vitro data in Fig 2 suggested that IFN-gamma production is either not affected or enhanced in ICER/CREM KO mice. However the data on the EAE model in Fig 5 demonstrates that IFN-gamma is suppressed in the ICER/CREM KO mice. This discrepancy needs to be explained.
- 3) The clinical disease in the EAE model is relatively modest. It would be good to repeat this with a more robust disease course (e.g using a higher dose of PT) to see if the reduction in diseases is still observed in the ICER/CREM KO mice.
- 4) The reduction in CD4 cells and IL-17A+ cells in the spinal cord of ICER/CREM KO mice is not significant when compared with the WT mice. The numbers of mice are small (4/group). This may need to be repeated to increase the n values and improve the strength of the data.
- 5) The IL-17 data in Fig 5G on the lymph node is not very convincing (about 2.5% to 1.5%). This would might be better if the cells were re-stimulated with MOG and IL-17A and IFN-gamma was quantified by ELISA
- 6) The authors claim that Th17 cells express ICER, whereas Th1 cells do not. This conclusion is not in line with the data shown in the sample immunoblot. There is a strong signal from the Th1 cells.
- 7) The crucial data in Fig 7C is weak for the conclusions made from it. There are only 3 control samples and only 3 of 6 samples from SLE patients show expression of ICER. The numbers of patients and controls needs to be increased. This might also be more convincing using Th17 cells (purified as for FIG 7B) rather than total CD4 cells.

Reviewer #2

Expert in glomerulonephritis

(Remarks to the Author):

This report shows that ICER is induced in Th17 cells through the IL-6/STAT3 pathway and that Its binding to the IL17a promotor results in the accumulation of enhancer ROR γ t. The role of ICER in Th17 induction was confirmed in vitro by rescuing Th17 induction of ICER/CREM deficient CD4 T cells by forced overexpression of ICER. In agreement with the role of Th17 in autoimmune disorders and autoinflammatory diseases, ICER/CREM deficiency ameliorates the development of anti-glomerular basement membrane-induced glomerulonephritis, experimental encephalomyelitis and the autoinflammatory disease of B6.lpr. In addition, over-expression of ICER was documented in lupus patients'sCD4+ cells. These findings support a unique role of ICER in Organ specific and systemic autoimmunity in a Th17 dependent manner.

The results are of significant interest to investigators in the field of autoimmunity and add significantly to our understanding of Th17 induction. The linkage of ICER to Th17 induction is novel. The data are of good quality with appropriate statistical analysis. The conclusions are in

agreement with the data. The multi approaches taken reassure us the reliability, validity of the conclusion.

BY large the presentation is concise and the references to prior work are adequate.

The manuscript would be improved by editing the summary with the order of the findings corresponds to the order of the figures as presented in the Result Section.

Reviewer #3

Expert in CREB/cAMP signalling

(Remarks to the Author):

This report shows that the cAMP early repressor, ICER, which is a CREM splice variant without the transactivating domain and that represses CREM/CREB signalling is induced during Th17 cell differentiation by IL6 signaling through Stat3. ICER appears to be regulate ROR γ t expression and to be required for Th17 differentiation as evident from experiments with T cells from CREM $^{-/-}$ mice whereas ICER re-expression in CREM $^{-/-}$ rescues Th17 differentiation. The authors go on to show that development of the autoimmune model diseases anti-glomerular basement membrane induced glomerulonephritis (AING) and experimental encephalomyelitis (AEA) are attenuated in CREM $^{-/-}$ in line with the role for Th17 in these models. Lastly the authors show that ICER is overexpressed systemic lupus erythematosus, SLE, humans.

This is a well-conducted paper in a topical and important area of immunology and inflammation is, to my knowledge, entirely novel. I have only a few comments:

- 1) It would be good to see Stat3 phosphorylation levels in Figs 3A and B.
- 2) Why is the statistical analysis in Fig 4C performed as a multiple comparison across all groups when for example panel B of that same figure has pair-wise comparisons? The authors could for example have decided to compare the ICER/CREM $^{-/-}$ to the $+/+$ at different time points.
- 3) Same in Fig 5C and O, why were group-wise statistics used here and not in other figures?
- 4) What signal does the ICER repressor interfere with in Th17 cells? Would this be a cAMP pathway (if yes, which one?) that signals through CREB/CREM or a signal through another kinase than PKA that can also phosphorylate CREB/CREM or act through the CRE. Good if the authors could address this point, or at least discuss what possibilities there are.

Point-by-Point Answers to the Reviewers' Comments

We deeply appreciate the constructive suggestions on improving our manuscript by three reviewers. As requested by the reviewers, we have performed additional experiments as described in detail below. With these extensive revisions, we believe that our paper has been substantially strengthened.

Reviewer #1

This study shows that the inducible cAMP early repressor (ICER) is involved in development of Th17 cells that are pathogenic in autoimmune diseases. The study is novel and the data is mostly convincing, especially in animal models with less extensive or convincing human data using PBMC from patients with SLE.

Specific comments

1) The data with in vitro polarized Th1, Th2, and Th17 cells is based on IL-6 and TFG-beta polarized Th17 cells from naive mice, activated with anti-CD3 and CD28. It is not clear whether ICER is induced by cell activation in vitro or expressed by Th17 cells in vivo. This would be enhanced with data on ex vivo Th17 from mice with autoimmune diseases either without re-stimulation or following activation with IL-1 and IL-23, the activation stimulus for memory Th17 cells.

The reviewer is raising a basic point. To determine whether ICER is present in *in vivo* cells prior to polarization, we purified IL-17 producing T cells from 18 weeks old diseased MRL/*lpr* mice and naïve CD4 T cells from 18 weeks old non-disease control MRL/Mpj mice, by flow cytometry. Using this method we could acquire about 1 million cells from approximately 2 billion (a frequency of approximately 0.05%!) spleen and lymph node cells. Since we had to stain for intracellular IL-17 in order to detect IL-17 producing T cells, we used a special buffer to acquire cell lysates from those samples. And as we show below, we recorded that ICER expression was increased in *in vivo* Th17 cells from diseased mice compared to naïve CD4 T cells from non-diseased control mice. We repeated this experiment 3 times using 3 different pairs of mice. We analyzed the density of the band using image J software and indeed ICER levels were increased ($p=0.03$) in non-induced Th17 cells from MRL/*lpr* mice. This figure is now shown in Figure1D.

2) The *in vitro* data in Fig 2 suggested that IFN-gamma production is either not affected or enhanced in ICER/CREM KO mice. However the data on the EAE model in Fig 5 demonstrates that IFN-gamma is suppressed in the ICER/CREM KO mice. This discrepancy needs to be explained.

The discrepancy between *in vitro* and *in vivo* comes from the difference of disease severity *in vivo* experiment between ICER/CREM deficient mice and sufficient counterparts. Since we calculate the number of infiltrated T cells based on the absolute number of cells in spinal code and the total absolute number of infiltrated T cells was decreased in spine from ICER/CREM deficient mice than that in spinal code from the counterparts. We have corrected the sentence as follows ‘Since the number of infiltrated T cells was reduced in spinal cord of ICER/CREM deficient mice, the absolute numbers

of CD4⁺ (CD45⁺CD90.2⁺CD4⁺; Figure 5D), IL-17A⁺ (CD45⁺CD90.2⁺CD4⁺IL-17A⁺; Figure 5E) and IFN γ -producing cells (CD45⁺CD90.2⁺CD4⁺IFN γ ⁺; Figure 5F) were reduced in the spinal cord from ICER/CREM deficient mice as compared to those in the spinal cord of ICER/CREM-sufficient counterparts and this reflects the decreased disease severity of ICER/CREM-deficient mice’.

3) *The clinical disease in the EAE model is relatively modest. It would be good to repeat this with a more robust disease course (e.g using a higher dose of PT) to see if the reduction in diseases is still observed in the ICER/CREM KO mice.*

Unfortunately, since our Institutional Animal Care and Use Committee (IACUC) did not allow us to challenge with higher doses of MOG and pertussis toxin, and since they expect us to terminate EAE experiments prior to the animals reaching the clinical score grade of 4, we cannot address this request.

4) *The reduction in CD4 cells and IL-17A⁺ cells in the spinal cord of ICER/CREM KO mice is not significant when compared with the WT mice. The numbers of mice are small (4/group). This may need to be repeated to increase the n values and improve the strength of the data.*

We have repeated this experiment and the *n* values was been increased (8/group) as shown below and in Figure 5 (D, E, and F).

5) *The IL-17 data in Fig 5G on the lymph node is not very convincing (about 2.5% to 1.5%). This would might be better if the cells were re-stimulated with MOG and IL-17A and IFN-gamma was quantified by ELISA*

As suggested, we performed a MOG re-stimulation experiment. EAE was induced in ICER/CREM deficient/sufficient mice (6/group each), and draining lymph nodes were harvested on day 8. Isolated cells from the lymph nodes were further cultured *ex vivo* with MOG for 3 days. IL-17A/F and IFN γ concentrations were measured by ELISA. As shown below and also in Fig. 4, IL-17A/F production was decreased in ICER/CREM deficient mice, whereas IFN γ production did not reach a significant difference.

6) The authors claim that Th17 cells express ICER, whereas Th1 cells do not. This conclusion is not in line with the data shown in the sample immunoblot. There is a strong signal from the Th1 cells.

The statement was based on the fact that only in Th17 cells ICER expression reached a statistically 'significant' difference (figure 7C). We have toned down our sentence and as follows 'ICER was present in significant amounts only in Th17 cells rather than any of the other T cell subsets.'

The crucial data in Fig 7C is weak for the conclusions made from it. There are only 3 control samples and only 3 of 6 samples from SLE patients show expression of ICER. The numbers of patients and controls needs to be increased. This might also be more convincing using Th17 cells (purified as for FIG 7B) rather than total CD4 cells.

We have expanded the *n* (SLE *n*=9, HC *n*=17) and ICER expression (ratio of ICER and actin) is shown in Figure 7C (right side). We have attempted to isolate Th17 cells by flow cytometry, however it was impossible to collect a sufficient number of Th17 cells to extract sufficient amounts of protein for Western blotting.

Reviewer #2

Expert in glomerulonephritis

(Remarks to the Author):

This report shows that ICER is induced in Th17 cells through the IL-6/STAT3 pathway and that its binding to the IL17 α promoter results in the accumulation of enhancer ROR γ t. The role of ICER in Th17 induction was confirmed in vitro by rescuing Th17 induction of ICER/CREM deficient CD4 T cells by forced overexpression of ICER. In agreement with the role of Th17 in autoimmune disorders and autoinflammatory diseases, ICER/CREM deficiency ameliorates the development of anti-glomerular basement membrane-induced glomerulonephritis, experimental encephalomyelitis and the autoinflammatory disease of B6.lpr. In addition, over-expression of ICER was documented in lupus patients' CD4⁺ cells. These findings support a unique role of ICER in organ specific and systemic autoimmunity in a Th17 dependent manner. The results are of significant interest to investigators in the field of autoimmunity and add significantly to our understanding of Th17 induction. The linkage of ICER to Th17 induction is novel. The data are of good quality with appropriate statistical analysis. The

conclusions are in agreement with the data. The multi approaches taken reassure us the reliability, validity of the conclusion.

BY large the presentation is concise and the references to prior work are adequate. The manuscript would be improved by editing the summary with the order of the findings corresponds to the order of the figures as presented in the Result Section.

We appreciate the reviewer's encouraging and supportive comments. We have edited the summary and improved the overall appearance of the manuscript.

Reviewer #3

Expert in CREB/cAMP signalling

(Remarks to the Author):

This report shows that the cAMP early repressor, ICER, which is a CREM splice variant without the transactivating domain and that represses CREM/CREB signalling is induced during Th17 cell differentiation by IL6 signaling through Stat3. ICER appears to be regulate RORgt expression and to be required for Th17 differentiation as evident from experiments with T cells from CREM^{-/-} mice whereas ICER re-expression in CREM^{-/-} rescues Th17 differentiation. The authors go on to show that development of the autoimmune model diseases anti-glomerular basement membrane induced glomerulonephritis (AING) and experimental encephalomyelitis (AEA) are attenuated in CREM^{-/-} in line with the role for Th17 in these models. Lastly the authors show that ICER is overexpressed systemic lupus erythematosus, SLE, humans.

This is a well-conducted paper in a topical and important area of immunology and inflammation is, to my knowledge, entirely novel. I have only a few comments:

1) *It would be good to see Stat3 phosphorylation levels in Figs 3A and B.*

We performed an experiment shown in Fig 3A and Fig 3B in which we checked Stat3 phosphorylation levels. Both Stat3 and phosphorylated Stat3 (pStat3) expression levels were added to the new figure 3A. For the figure 3B, as shown in below, since STA-21 inhibits Stat3 DNA-binding activity and Stat3 dimerization and doesn't inhibit the phosphorylation of Stat3 levels, we could not detect any reduction of pStat3 in the presence of STA21.

2) *Why is the statistical analysis in Fig 4C performed as a multiple comparison across all groups when for example panel B of that same figure has pair-wise comparisons? The*

authors could for example have decided to compare the ICER/CREM -/- to the +/+ at different time points.

3) Same in Fig 5C and O, why were group-wise statistics used here and not in other figures?

We present the significance across all groups because we used 2way ANOVA to analyze the data. We also applied Bonferroni's multiple comparisons test, and found that on the 21day of the amount of proteinuria was significant in Figure as well as additional points. Because it becomes too busy to show all the significant points, we chose to keep the original presentation. We mention the statistical method in the figure legend.

4) What signal does the ICER repressor interfere with in Th17 cells? Would this be a cAMP pathway (if yes, which one?) that signals through CREB/CREM or a signal through another kinase than PKA that can also phosphorylate CREB/CREM or act through the CRE. Good if the authors could address this point, or at least discuss what possibilities there are.

We added the following sentences. 'Since it has been reported that levels of cAMP are increased in ICER/CREM deficient mice, cAMP is known to regulate glycolysis and glycolysis plays a critical role in Th17 cell differentiation, we speculate that ICER/CREM affects the glycolysis pathway'.

REVIEWERS' COMMENTS:

Reviewer #1 (Remarks to the Author):

The authors have addressed most of my comments satisfactorily and the manuscript is significantly improved.

Reviewer #3 (Remarks to the Author):

I have read the revised manuscript and comments by the authors to the reviewers' questions. The responses and additional experiments have in my view addressed the comments and concerns raised.

Minor: in new Fig 1D why is the order of the two samples inverted in the graph compared to the blot to the left? In my view it would be clearer if the order of the samples were the same.

Point-by-Point Answers to the Reviewers' Comments

We deeply appreciate the constructive suggestions on improving our manuscript by reviewers. As requested by the reviewers, we have edited our figure as described in detail below. We believe that our paper has been substantially strengthened.

Reviewer #1

The authors have addressed most of my comments satisfactorily and the manuscript is significantly improved.

We appreciate the reviewer's encouraging and supportive comments.

Reviewer #3 (Remarks to the Author):

I have read the revised manuscript and comments by the authors to the reviewers' questions. The responses and additional experiments have in my view addressed the comments and concerns raised.

Minor: in new Fig 1D why is the order of the two samples invested in the graph compared to to the blot to the left? In my view it would be clearer if the order of the samples were the same.

We appreciate the reviewer's encouraging and supportive comments. In Figure 1D, we have re-ordered the samples in the graph in the same order as in the blot.